# Immune Response against Adenovirus in Acute Upper Respiratory Tract Infections in Immunocompetent Children

**DOI:** 10.3390/vaccines8040602

**Published:** 2020-10-13

**Authors:** Giovanni Battista Biserni, Arianna Dondi, Riccardo Masetti, Jessica Bandini, Ada Dormi, Francesca Conti, Andrea Pession, Marcello Lanari

**Affiliations:** 1Pediatric Emergency Unit, Scientific Institute for Research and Healthcare (IRCCS), Sant’Orsola Hospital, 40138 Bologna, Italy; giovanni.biserni@studio.unibo.it (G.B.B.); marcello.lanari@unibo.it (M.L.); 2Pediatric Unit, Scientific Institute for Research and Healthcare (IRCCS), Sant’Orsola Hospital, 40138 Bologna, Italy; riccardo.masetti@gmail.com (R.M.); jeba89@alice.it (J.B.); francescac.ageop@aosp.bo.it (F.C.); andrea.pession@unibo.it (A.P.); 3Department of Medical and Surgical Sciences, DIMEC, University of Bologna, 40138 Bologna, Italy; ada.dormi@unibo.it

**Keywords:** Adenovirus, cytokine, immune system, lymphocytes populations subsets, acute upper respiratory tract infections, children

## Abstract

During acute upper respiratory tract infections (AURTIs) caused by Adenoviruses, the mix of severe clinical presentation, together with elevation of white blood cells (WBCs) and C-reactive protein (CRP), often mimicking bacterial infection, leads to an inappropriate use of antibiotics. We studied 23 immunocompetent children admitted to our Pediatric Emergency Unit with signs of acute Adenoviral AURTIs, aiming at better clarifying the biological background sustaining this clinical presentation. Infection etiology was tested with nasopharyngeal swabs, serology, and DNA-PCR. During fever peaks and subsequent recovery, we assessed WBC count with differential, CRP, procalcitonin, serum concentration of six inflammatory cytokines, and lymphocyte subset populations. Results: IL-6 and IL-8 were found elevated in the acute phase, whereas a significant decrease during recovery was found for IL-6 and IL-10. We highlighted an increase of B lymphocytes in the acute phase; conversely, during recovery, an increase in T regulatory cells was noted. Monocytes and leukocytes were found markedly elevated during fever peaks compared to convalescence. All patients recovered uneventfully. The composition of lymphocyte population subsets and serum alterations are the main drivers of an overprescribed antibiotic. Examination of hospital admissions and performance is needed in further investigations to rule out bacterial infections or inflammatory syndromes.

## 1. Introduction

Human Adenoviruses (HAdVs) are double-stranded, nonenveloped DNA viruses belonging to the genus Mastadenovirus of the *Adenoviridae* family [1,2,3], known to be potentially associated with strong inflammatory responses and morbidity in pediatric patients. At present, more than 100 different types of HAdV have been described and are designated with consecutive numbers [4,5]. HAdVs might be found as the causative agents of gastroenteritis, hepatitis, encephalitis, conjunctivitis, coryza, pharyngitis, and ear inflammation. Though most of the primary infections are self-limiting in the immunocompetent host, the severity of clinical presentation combined with the elevation of the white blood cell (WBC) count and C-reactive protein (CRP), mimicking bacterial infection, often lead to hospital admission and inappropriate use of antibiotics. Inappropriate prescriptions of antibiotics, which frequently occurs in acute upper respiratory tract infections (AURTIs) [6], is the main contributor to antimicrobial resistance, and may lead to an unacceptable economic burden, not mentioning the increase incidence of side effects in patients [7]. Moreover, the clinical presentation may mimic other illnesses such as Kawasaki disease and thus requires further investigations, increased parental care, anxiety, and loss of working days resulting in expenses for both the health and economic systems [8].

### 1.1. Clinical Features

Most infections occur via aerosolized droplets or direct conjunctival inoculation, but the fecal–oral route or contact with contaminated fomites are also possible [3,9]. The incubation period varies from 2 days to 2 weeks depending on virus type and on the transmission route [10].

HAdV can cause highly different systemic pictures, according to the age group, community settings, immune status, and HAdV species [3]. In Table 1, common associations between infections sites and HAdV species are reported. In younger children, acute upper respiratory tract infections (AURTIs) are the most common clinical manifestations; lower respiratory tract infections and gastrointestinal (GI) symptoms are also frequent in this age group and may be severe yet, rarely, lethal [11]. Moreover, AURTIs and GI symptoms often coexist [2]. In older children, the spectrum of HAdV disease includes AURTIS, pharyngoconjunctivitis, non-purulent swimming pool conjunctivitis, hemorrhagic cystitis, and mesenteric adenitis [3]. In the pediatric age, HAdV respiratory illnesses may be complicated by the persistence of the infection that may elicit chronic neutrophilic inflammation in the airways, protracted bacterial bronchitis and bronchiectasis [12]. Long-term respiratory sequelae include bronchiolitis obliterans, HAdV being the most frequent virus involved in its pathogenesis [13]. Moreover, neurologic complications have also been described in previously healthy children, with manifestations including febrile seizures, encephalitis, acute disseminated encephalomyelitis, and aseptic meningitis [14].

Any condition determining a severe immunosuppression with a lack of cellular activity confers a high risk of invasive and disseminated disease. In particular, patients undergoing allogeneic hematopoietic stem cell transplantation have a greater risk of Adenovirus related complications, resulting from de novo infection or reactivation of persistent endogenous virus [15]. In recipients of hematopoietic stem cell transplantation, HAdV infections have an incidence of up to 21% and are 2–3.5 times more likely in children compared with adults [2,15]. The frequency of invasive disease is 8–26%, and, in the case of respiratory tract infections, dissemination and/or severe respiratory failure develops in 10–30% of cases with a fatality rate possible of exceeding 50% [2,15]. Other conditions predisposing patients to more serious clinical conditions related to HAdV infections are being a recipient of solid organ transplantation, severe combined immunodeficiency syndrome, and acquired immunodeficiencies such as Human Immunodeficiency Virus infection [3,16].

### 1.2. Immune Response

HAdV disease is controlled by both innate and adaptive immune responses. The infection promptly elicits secretion of gamma interferon (IFN-γ), tumor necrosis factor (TNF), interleukin-1 (IL-1), IL-2, and macrophage inflammatory protein; these cytokines have an antiviral activity and limit the amplification and spread of the virus [9]. At the same time, natural killer cells are recruited and activated to destroy the cells infected by the virus [17]. On the other hand, viral products mediate the transcriptional repression of IFN-inducible genes and this process facilitates viral replication [18].

On in vitro models, when HAdV-specific T cells are produced, they are able to induce lysis of infected cells by a perforin-dependent mechanism [19]. HAdV-specific CD4 and CD8 T cell subsets were shown to display cross-reactivity with different adenoviral species and types [20,21]. It has been hypothesized that HAdV immunity in adults is carried by cross-reactive cytotoxic T cells generated by exposure during infancy and childhood, and, thanks to their cross-reactivity with different adenoviral types, can be protective against most or possibly all HAdV forms [9]. Previous studies focused on HAdVs ability to escape interferon-induced transcription in the hosting cells and promote viral replication [18], others discovered that different HAdV species and variations in the whole genome sequencing of the host may cause more severe respiratory disease in children [22,23].

### 1.3. Aim of This Study

The vast majority of the studies on HAdV immune response were conducted in vitro or in vivo on models not including human subjects. The immune system plays a key role in the mechanisms that lead to severe presentation, but, to date, the molecular bases of that mechanism are poorly understood. To address this issue, this cross-sectional study aimed at verifying the pattern of the immune response to HAdV in terms of inflammatory cytokines and lymphocytes population in pediatric patients affected by AURTIs requiring hospitalization.

## 2. Materials and Methods

### 2.1. Study Design and Participants

The target population comprised all pediatric patients <14 years old admitted to the Pediatric Emergency Department of Sant’Orsola University Hospital in Bologna, Italy, with a diagnosis of AURTIs such as pharyngitis, tonsillitis, coryza, or ear inflammation. Patients were enrolled from April 2018 to April 2020. Inclusion criteria were as follows: presence of clinical signs and/or symptoms of upper respiratory tract infection at physical examination, fever >38.5 °C, parental informed consent. Exclusion criteria comprised any of the followings: comorbidities such as complex cardiopathies, chronic or end stage renal failure, primary immune system deficiency, solid tumors or malignant hematological disease, as well as stem cell transplantation recipients, and subjects chronically treated with biological and immunomodulating therapies, and patients with signs of localization of infection other than upper respiratory tract at admission. Criteria for hospital admission were the need for supportive care in order to maintain adequate hydration, and the presence of an ill-appearing child requiring differential diagnosis with serious bacterial or inflammatory disease (such as sepsis or Kawasaki disease), according to clinician’s judgment.

### 2.2. Procedures and Measurements

Upon admission, beside the routine blood tests (WBC and differential adjusted for age, CRP, procalcitonin), six inflammatory cytokines, naso-pharyngeal swabs for adenoviral antigen, HAdV serology, and HAdV DNA-PCR on blood or on target fluids were obtained. The most common agents causing AURTIs in children (RSV, Influenza virus, Parainfluenza virus, Human Metapneumovirus, CMV, EBV, Chlamydia pneumoniae, Mycoplasma pneumoniae, Streptococcus Pyogenes) were also ruled out by antigen detection on naso-pharyngeal swab, serologic tests, and rapid film-array PCR. A supplemental 1-mL blood sample was obtained to assess the composition of lymphocytes population subsets. As we do not use rapid antigen detection (RAD) kits, results took 24–48 h depending on the method.

Adenoviral AURTI was defined by positivity of either antigen detection, serum IgM, or PCR-DNA in blood or body fluids. Based on etiological findings, the sample was then divided into positive or negative patients for adenoviral infections. Patients positive for other respiratory pathogens were excluded, to avoid confounding results. The former subgroup had routine blood tests repeated, (including serum cytokines and the lymphocytes) once clinically improved, meaning that they were afebrile for at least 24 h.

For ethical reasons, we did not perform additional phlebotomy for the purposes of the study other than the one included in our standard practice. For this reason, in the acute phase of the infection, we collected blood samples of 23 patients, lymphocytes populations were assessed in 18 patients, inflammatory cytokines in 14 patients; during convalescence samples were collected from 19 of the patients, lymphocytes populations were assessed in 16 patients, inflammatory cytokines in 12 patients.

### 2.3. Inflammatory Cytokines

We assessed the serum concentration of six different cytokines (IL-6, IL-8, IL-12p70, IL-1β, IL-10, TNF-α), which were analyzed according to the protocol adopted by the Unified Metropolitan Laboratory of Bologna employing sandwich chemiluminescence immunoassay (SCIA) for IL-6, and cytofluorometry for the others. The results were compared to normal values for age, when available [24]. 

### 2.4. Identification of Lymphocyte Populations

The lymphocyte populations of the host were analyzed based on surface antigen detected by Flow Cytometry. The results were interpreted with a six-color Cytometer FACSCanto II (BD, Becton Dickinson). Blood samples were divided in three test tubes (Falcon Round-Bottom Tube BD Becton Dickinson). In each tube, 100 μL was added to a specific antibody panel from BD Biosciences (BD, Becton Dickinson), as follows:

- anti-CD3 FITC (T lymph), anti-CD16 PE (NK lymph), anti-CD56 PE (NK lymph), anti-CD20 PERCP (B lymph), anti-CD14 APC (monocytes), anti-CD45 APC-H7 (Leukocytes).

- anti-CD3 FITC (T lymph), anti-CD25 PE (Reg T), anti-CD127 PERCP (Reg T), anti-CD8 PECY7 (T lymph cytotoxic), anti-CD4 APC (T lymph helper), anti-CD45 APC-H7 (Leukocytes).

- anti-CD45RA FITC (T lymph naive), anti-CD45R0 PE (T lymph memory), anti-CD3 PERCP, anti-CD8 PECY7 (T lymph cytotoxic), anti-CD4 APC (T lymph helper), anti-CD45 APC-H7 (Leukocytes).

Each tube was sealed as indicated by the manufacturer and incubated for 15 min at room temperature. After that, 2.5 mL of sterile ammonium chloride solution ph 7.4 and Pasteur solution were added to achieve red cell lysis. The solution acted for 10 min at room temperature inside of each sealed tube. The lysates were then centrifuged at 1500 rpm for 5 min and the precipitates were taken and reconstituted in 500 μL of PBS (phosphate buffered saline, Sigma-Aldrich). According to the manufacturer pre-setting, the Cytometer FACSCanto II ran the analysis and the software BD FACSDiva (BD, Becton Dickinson, Milan, Italy), allowed the acquisition of at least 80,000 events from each sample (gates of lymphocyte), on which the subset of lymphocyte populations was calculated in percentage.

### 2.5. Sample Size Calculation and Statistical Analysis

In order to detect a clinically significant difference in mean between groups of greater than 5% with a power of 80% (type II error β = 0.2) and an α value of 0.05 (type I error), correlation within pairs of 0.60, and to account for protocol failure (up to 20%), the necessary sample size was calculated as 16 patients.

Normality was tested according to Kolmogorov–Smirnoff. Categorical variables were presented as frequencies or percentages; continuous variables with normal distribution were presented as the mean values, standard deviations, and confidence intervals; continuous variables, not normally distributed, were presented as median and interquartile range. Differences in sample means were compared using a paired two-tailed Student’s test or Wilcoxon test, when appropriate. 

Levels of IL-6 and monocytes were compared with linear regression, in both phases. Difference in linear regressions were compared with Line regression t-test. Missing data were corrected pairwise. All analyses were conducted with IBM SPSS Statistics for Windows, 22.0 or following versions (IBM Corp., Armonk, NY, USA).

### 2.6. Ethics

The study protocol was approved by the Ethical Committee of Sant’Orsola University Hospital (Area Vasta Emilia Centro, protocol n. 391/2019/Sper/AOUBo). Written informed consent was obtained by the parents or the caregivers.

## 3. Results

### 3.1. Population Characteristics

Demographic and clinical data of the population are reported in Table 2. Of the 48 eligible participants, 23 (46.1%) showed AURTIs in which HAdV was found as the etiological agent. Male to female ratio was 1.8:1 and the mean age was 2.2 ranging from 7 months to 7 years. The length of hospitalization ranged from 2 to 6 days. The majority of patients received antibiotics. Antibiotics were mostly given empirically before admission after consultation with the GP or pediatrician, neither of whom employ RAD routinely. In our patients, antibiotic therapy was continued if started at home, reassessed on the third day of fever, and discontinued as microbiological evidence of HAdV was available. When started in-hospital, antibiotics were given empirically on the evidence of an ill-appearing child with elevated WBC count with neutrophilic differential, and elevation of CRP and/or Procalcitonin, after cultures appropriate to pediatricians’ discretion were obtained. No patient had a chronic illness or evidence of immunosuppression. No patients needed oxygen therapy and the three cases of associated pneumonias were mild. The majority of HAdV AURTIs patients were hospitalized during the spring or summer months (60.1%). With regards to seasonal distribution, it should be noted that the national lockdown, set up during the late winter months in year 2020 to limit the spread of the SARS-CoV2 pandemics, with early school and kindergarten closures, may have brought about a reduction in the circulation of Adenovirus at once [25].

As regards clinical outcome, all patients recovered completely.

### 3.2. WBC Count

We analyzed WBC count and differential of the 23 AURTIs caused by Adenovirus. No patient showed lymphopenia or neutropenia according to age limits. The two phases of the disease differed both in percentage and in absolute count for most of the cell populations. The acute phase of the Adenoviral infection was found to be dominated by neutrophilic leukocytosis, with an average of approximately 13,000 neutrophils/μL, whereas convalescence showed a return to a lymphocyte-predominant differential, which is appropriate for age in normal conditions. The analysis was performed to confirm the shift to a less inflammatory status, reflected by the abrupt decrease in CRP and procalcitonin with clinical improvement and afebrile status (Table 3).

### 3.3. Lymphocytes Subsets

The subsets of lymphocyte populations showed an increase of B lymphocytes both in percentage and absolute count in the acute phase; conversely, during convalescence, an increase in T regulatory (T reg) cells was noted compared to the acute phase of the infection (Table 4).

### 3.4. Serum Cytokines

In the acute phase, serum cytokines were found markedly elevated from normal values in cases of IL-6 and IL-8. Differences in concentration between the two phases were found for IL-6 and IL-10, both showing a statistically significant decrease during convalescence (Table 5, Figure 1). The regressions between IL-6 and monocytes differed between the two phases (*p* = 0.035).

## 4. Discussion

### 4.1. General Considerations

This study shows that HAdV elicits a strong systemic inflammation in pediatric AURTIs. In parallel with previous studies [26,27], it also shows a marked increase in CRP and procalcitonin together with WBC and neutrophilic profile promptly normalizing once the acute phase is over. Even in the setting of mild AURTIs, systemic inflammation with alterations of serum inflammatory markers have already been established [23,26]. Serum concentration of IL-6, IL-1β, TNFα, IFNγ were found elevated in cases of HAdV, RSV, and influenza AURTIs compared to age-specific normal values. CRP, whose production by the liver is driven by IL-6, showed a difference between HAdV and other pathogens and a positive correlation with IL-6 [26]. Procalcitonin is readily released by parenchymal cells and immune cells in response to inflammation [28]. It has been widely used to guide antimicrobial prescriptions both in in-hospital and outpatient settings amongst adult patients with lower respiratory tract infections. Evidence regarding the role of procalcitonin in children in AURTIs is lacking, although it has been observed that in case of respiratory tract infections caused by HAdV it does not differ between AURTI and pneumonia and was within the normal range for adults [27]. The serum alterations found in this study, together with clinical manifestations that may mimic a serious bacterial infection, lead to hospital admissions, investigations, and inappropriate antibiotic prescribing. 

In this study in children without comorbidities, fever resumed on average between the second and third day of hospital stay; concomitantly, clinical conditions improved dramatically. The self-limiting nature of the disease, together with the irrelevance of antimicrobial therapy on its course, either orally or intravenously, calls for tools in order to promptly diagnose a HAdV etiology in case of AURTIs with systemic symptoms. Rapid antigen detection kits for nasopharyngeal swabs can represent suitable tools even in outpatients; in cases of prolonged fever in a child that does not appear seriously ill, well hydrated and with no clear signs of a bacterial infection (including streptococcal tonsillitis), the identification of HAdV in secretions may avoid ineffective and potentially dangerous treatment and management strategies [29]. Indeed, a rapid HAdV detection would prevent the administration of useless antibiotic therapies with potentially harmful toxicity and increase in antimicrobial resistance, and the exposure of children to stressful situations or environment beyond the disease itself (blood tests, chest X-ray, hospital stay). Moreover, the use of antimicrobials in children is particularly controversial, given that fewer molecules have been safely tested in younger patients compared to adults, and that the employment of a drug is often empirical. Unfortunately, most of the existing rapid diagnostic kits are intended for research purposes only and few of them have been efficaciously tested in real-life pediatric medicine [29].

### 4.2. WBC Count and Lymphocyte Populations Subsets

The groups of WBC that showed the major variations between the acute phase and convalescence were monocytes and neutrophils, whose absolute counts were reduced by 61.5% and 68.8%, respectively. We encountered a decreased number of eosinophils both in percentage and absolute count during the acute phase. In animal models, eosinophils are involved in cytokines secretions (among which IL-6, TNF-α, IL-12p70) that are both preformed in granuli and can be readily expressed in response to respiratory viruses; moreover, eosinophils display the role of clearing viral particles in respiratory syncytial virus and parainfluenza virus infections and serve as antigen presenting cells for TCD4+ lymphocytes in cases of rhinovirus [30,31]. These actions are accomplished locally, in the respiratory epithelium of the affected model. Appropriately, in our study we found a decrease in eosinophils during the acute phase that may reflect the recruitment of these granulocytes in the upper respiratory tract during the early phases of the infection [32]. 

HAdV can be readily internalized by tissue macrophages and generates a rapid inflammatory response. To date, at least three different patterns of immune response are responsible for monocyte-derived macrophage activation, cytokine secretion, and inflammation. In murine respiratory epithelium, macrophages are the main findings in bronchoalveolar lavage of mice infected with Adenovirus and are encountered in every time point of the assessment up to 6 h from instillation [33]. Alveolar macrophages internalize Adenovirus through CAR (Coxsackie Adenovirus Receptor), MARCO (Macrophage receptor with Collagenous structure), and other toll-like and scavenger receptors (TLR-2,4,9). Macrophages do not strongly acidify the lysosomes and provide a favorable environment for both genome and viral particles to escape neutralization especially for viruses that do not need acidification, such as Adenovirus [34]. This triggers intracellular proinflammatory signaling through NF-kB, IRF-3, and p38 to induce cytokines secretion with high levels of IL-6 and IL-1α, occurring eventually in cell death [35]. High concentration of IL-6, TNF-α, and IL-8 were found elevated in other cases of respiratory tract infections in different clinical settings in children (outpatients, inpatients, and intensive care units) [26,36].

HAdV was shown to cause activation of the inflammasome-mediated cell death in cell lines of monocyte-derived macrophages [34]. Even though macrophages are designed to suppress infections, their first-line role in sensing pathogens puts them in danger of being infected themselves [37]. Inflammasomes are complexes of proteins whose polymerization is induced by signals of danger in the cytosols (e.g., viral genome and fragments, intracellular cGAS receptor, NLRP3). Inflammasome induces a programmed cell death with release of IL-1 β, IL-18 [34]. The same receptors that mediate internalization of HAdV in alveolar macrophages are involved in this process, but recently, evidence points to the role of HAdV specific antibodies [34]. In the presence of high levels of antibodies, the classic phagocytosis response captures large amounts of HAdV particles, this results in lysosomal damage, NLRP3 activation, and cleavage of IL-1β by caspase [38]. In the presence of low levels of HAdV antibodies, macrophages were shown to drive secretion of proinflammatory molecules IL-1β and TNF-α without lysosomal damage, inflammasome activation, or cell death [34].

Finally, MARCO receptor is abundant on Kupffer cells and spleen macrophages [39]. MARCO has been shown to sequester particles of adenoviral vectors from the blood stream [39,40]. In mice, internalization of HAdV in Kupffer cells and the sensing of viral particles in the cytosol elicit necrotic cell death with release of high levels of inflammatory cytokines [40]. In the spleen, MARCO/HAdV infected macrophage in the marginal zone has been recently discovered to be the trigger of local cytokines production (IL-1α) and neutrophil chemotaxis, directly from bone marrow [41].

The elevation in monocytes observed in the present study, together with the parallel increase of IL-6 in the acute phase of the infection and the concurrent decrease during recovery, point towards the hypothesis that the inflammatory response might be sustained by the monocytes recruited at the site of infection. Our observation sustains the hypothesis that the production and release of IL-1 may represent only a local phenomenon throughout adenoviral AURTIs due to the low levels of IL-1β found in both phases. As with other stem cell precursors in mice [42], macrophage and neutrophils may be recruited from bone marrow due to early cell depletion mediated by cell death driven either by inflammasome, NF-kB, or “defensive death”.

Concerning adaptive immunity and lymphocyte population, we found differences in B lymphocytes and T regulatory cells between the two phases of the infection. Adenovirus is known to induce in vivo polyclonal activation of B lymphocytes [43]. On the murine model, the induction of the B cells proliferation started on day 4 and peaked at day 10 from infection. In our study, we found a decrease in B lymphocytes, on average, from the 6th to the 8th day from the onset of fever; considering an incubation period from 2 days to 2 weeks, our study found a similar peak to the murine model. In vitro studies showed that naïve B cells can be effectively transduced by HAdV. Recombinant HAdV of species 5 is internalized in endosomes, bound to TLR-9, and is able to activate NF-kB (a key mediator of TRL-9 pathway) and the secretion of proinflammatory cytokines (IL-1β) [44]. Furthermore, although in their naïve stage they harbor a limited ability to present antigen, B lymphocytes can readily express MHC I molecules in response to HAdV [45]. These can recruit and activate TCD8+ and induce degranulation and cytotoxic activity. Moreover, other cell types accompanying B lymphocyte increase, such as T helper, were found in accordance to previous results [43]. These latter cells showed little influence on the proliferation of B lymphocytes, but were fundamental in order to induce antibody production. 

In light of previous results [34,44], antibodies are shown to sustain inflammation in response to HAdV. Our serum samples were collected on mean day 3.2 after the onset of illness, when titers of adenoviral antibodies were already detectable. As our population included immunocompetent and healthy children, we cannot exclude that the presence of antibodies may represent a drive of immune mediated pathology in mild adenoviral AURTIs [34]. Notably, some cytokines may have peaked earlier in the course of adenoviral infection and differentiating between species of HAdV may have uncovered different immune response patterns. To complete these findings, total and anti-HAdV phase-specific antibody titers should be addressed in future studies.

### 4.3. Inflammatory Cytokines

No previous reports evaluated the role of IL-10 during HAdV infection in an inpatient setting. IL-10 is secreted during the early phases of infections by cells of the innate immune response (NK, Mast cells, APC, and granulocytes) and nonimmune cells types (fibroblasts, keratinocytes) in order to regulate local immune mediated damage. Lately during viral infections, T lymphocytes become the major source of this cytokine (T Helper, T CD8+, T reg) and are implicated in various activities such as viral clearance, antibody production, and fine tuning of local inflammation. Deficiency in IL-10 production can lead to the development of immunopathology by increased secretion of inflammatory mediators by Th1 [46]. During the resolution of viral infections and inflammation, T reg cells tower over other cell types in IL-10 secretion [47]. In this study, despite differing between the febrile phase and convalescence, the levels of IL-10 were within the normal range during adenoviral AURTIs, suggesting that, in these infections, this cytokine does not play a significant role in modulating systemic inflammation. Moreover, the increase in T reg, observed during convalescence, is not accompanied by IL-10 elevation, suggesting that these cells are not the source of serum IL-10 in the late phases of the infection.

IL-6 is produced by a variety of cells such as vascular endothelium, mononuclear phagocytes, fibroblasts, and activated T lymphocytes. In our study, linear regressions between the concentration of IL-6 and the number of monocytes in both phases suggest that the activation on macrophages at the site of infection could represent the main source of that cytokine. IL-6 acts as a stimulus for myelopoiesis and migration of neutrophils directly involved in inflammation and cytokine production [48]. In light of the abrupt decrease during convalescence, our study aims to highlight the role of this cytokine even in mild adenoviral infections. Similar differences were noted in B cells, of which IL-6 represent a differentiation factor and a stimulus for antibody productions.

Interestingly, studies focusing on inflammatory markers in order to discriminate between viral and bacterial respiratory tract infections, found an increase in IL-6 and procalcitonin in lower respiratory infections with bacteremia, and an overall tendency to higher levels of IL-1β, TNFα, IL-10, and IL-1ra in case of streptococcal pneumonia compared to influenza virus, RSV, and human metapneumovirus [49]. Mycoplasma pneumonia and streptococcal lower respiratory tract infections showed elevated levels of IL-6 compared to RSV [50].

In another study, the profile exhibited during the acute phase of adenovirus-induced AURTIs, with significant elevation of both IL-6 and IL-10, resembles the one of peripheral mononuclear cells stimulated with bacterial lysates; the production of these mediators seemed to be driven by B cells through TLR-2 pathway [51]. Immune stimulation of B cells in vitro with bacterial debrides elicits a response in the supernatant similar to what we encountered in the serum of our population during the acute phase, when IL-6, IL-10, and B cells were higher than during convalescence [51]. Although bacterial and viral infections have showed to differ significantly in terms of serum markers, taken together, previous experiences and our study, enforce the long-time experience that depicts HAdV as a viral agent with a bacterial “signature”.

The subsets of B cells that are the actual source of these cytokines have to be investigated, and the contribution of the single blood cell subpopulation is beyond the purpose of this work [51].

Finally, the secretion of IL-8, a neutrophilic chemotactic agent, elevated in both phases of the course, may explain the shift of WBC differential to neutrophilic predominant. 

### 4.4. Limitations of This Study

To the best of our knowledge this is the first study that focuses on lymphocyte population subsets and inflammatory cytokines in patients with Adenoviral AURTIs. However, it has some limitations. First of all, there are variations between the number of samples included in the analysis. Although we enrolled 23 patients in order to obtain an 80% statistical power, we were able to assess cytokines in a smaller proportion of the cohort. However, we addressed the missing data with legitimate statistical corrections. Secondly, only young, healthy children affected by HAdV AURTIs were examined, so the results might not be generalizable to other age and patient groups. This study is not able to draw conclusions on differences among various HAdV subtypes. The analyses on neutrophil nuclei would have added important information, however, we focused more on characterizing other immunological aspect extensively. Finally, another limitation of the study is the lack of a control group, which might be useful to check differences between immune response to HAdV and other viral or bacterial AURTIs and give a better insight on the actual role of the highlighted variations.

## 5. Conclusions

In immunocompetent children, HAdV is able to activate a strong systemic inflammation even in the case of AURTIs. In the acute phase of the infection, both innate and adaptive immunity are significantly triggered. The composition of lymphocyte population subsets and serum alterations, which are induced by adenoviral infections even in mild cases, are the main drivers of an abuse in antibiotic prescriptions, hospital admissions, and need for further investigations to rule out serious bacterial infections or inflammatory syndromes such as Kawasaki disease. Assessing the profile of inflammatory cytokines and lymphocyte populations broadens the knowledge of the inflammatory biological background of an Adenovirus infection in children leading to its better clinical management. This work might help for the identification of markers (IL-6, IL-8, IL-10, PCR, neutrophilic predominance, monocytes, or a combination of those) that triggers a prompt recognition of HAdV-related pathology and clinical pictures. Moreover, parents’ awareness should be raised about viral entities, such as HAdV, that may cause longer fever duration than expected, but in most cases are self-limiting.

## Figures and Tables

**Figure 1 vaccines-08-00602-f001:**
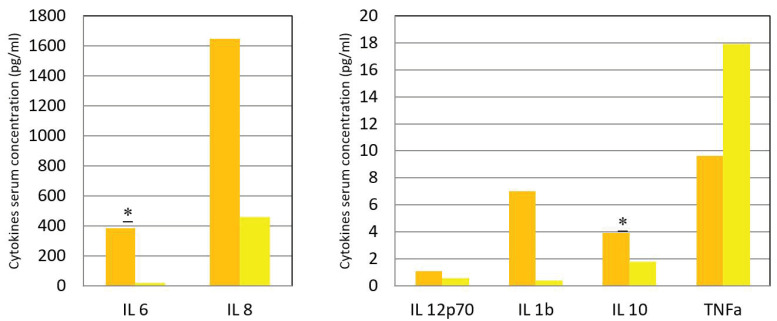
Inflammatory cytokines during the febrile (orange column) and afebrile (yellow column) phases. In ordinates, serum concentration of inflammatory cytokines in pg/mL. Asterisks represent significance at a *p* = 0.05 for IL-6 and *p* = 0.007 for IL-10, according to Wilcoxon sign-rank test.

**Table 1 vaccines-08-00602-t001:** Common associations between general infection sites and Adenovirus (HAdV) species. Other HAdV species may also occur at the site of infection. Many HAdV species show great variability in their tissue tropism.

Most Common Infection Sites	HAdV Species
Gastroenteritis	F and G
Pneumonia	B, C, and E
Hepatitis	C
Meningoencephalitis	A, B, and D
Cystitis	B
Keratoconjunctivitis	B and D

**Table 2 vaccines-08-00602-t002:** Demographic and clinical characteristics of the population.

Characteristic	Adenoviral AURTI
Gender, Male/Female	15/8
Age, mean (SD)	2.2 years (1.594)
Days of fever, mean (SD)	6.2 days (2.73)
Days of hospitalization, mean (SD)	3.3 days (1.3)
Physical findings (number of patients)	Nasal discharge (22);
	Pharyngitis (2019);
	Tonsillitis (15);
	Otitis (2);
	Bronchitis/Pneumonia (3);
	Conjunctivitis (1)
Method for diagnosis (number of patients)	Adenoviral Antigen (14);
	DNA PCR (110);
	Serology (6)
Coinfections	None

**Table 3 vaccines-08-00602-t003:** White blood cell (WBC) count, C-reactive protein (CRP), and Procalcitonin levels in 23 patients affected by Adenoviral AURTIs during the febrile (acute) and afebrile (convalescence) phases, *p* values reflect difference in means.

Test	Acute Phase	CI or IR	Convalescence	CI or IR	*p*
WBC (/µL)	12,880	10,870–16,990 *	8144	6854–9435	0.002
Neutrophils (/µL)	7464	5522–9406	1870	1430–3170 *	0.006
(%)	55.6%	49.9–61.4%	29.6%	24.9–34.4%	0.00
Lymphocytes (/µL)	5159	3916–6402	4898	4053–5743	0.66
(%)	34.6%	28.9–40.2%	60.1%	54.8–66.4%	0.00
Monocytes (/µL)	1241	941–1369 *	554	373–735	0.005
(%)	9.5%	8.1–10.8%	6.4%	4.9–7.9%	0.008
Eosinophils (/µL)	30	10–120 *	210	55–485 *	0.003
(%)	0.2	0–1.3% *	2.9	3.3–5.6% *	0.001
Basophils (/µL)	20	10–40 *	27	17–37	0.86
(%)	0.2	0.1–0.3% *	0.3%	0.2–0.4%	0.035
CRP (mg/dL)	8.3	5.9–10.6	2.9	2–3.8	0.000
Procalcitonin (ng/mL)	1.2	0.3–3.15 *	0.7	0.4–1	0.016

* Data not normally distributed.

**Table 4 vaccines-08-00602-t004:** Lymphocytes population expressed in concentration and percentage of total lymphocytes in 23 patients affected by Adenoviral AURTIs during the febrile (acute) and afebrile (convalescence) phases, *p* values reflect difference in means.

Test	Acute Phase	CI or IR	Convalescence	CI or IR	*p*
Total lymphocytes (/µL)	5159	3916–6402	4898	4053–5743	0.66
(%)	34.6%	28.9–40.2%	60.1%	54.8–66.4%	0.00
Natural killer (/µL)	414	235–562 *	385	234–536	0.6782
(%)	9.9%	7.3–12.5%	6.6%	4.2–9%	0.3281
B lymphocytes (/µL)	782	439–1125	754	555–953	0.011
(%)	25.1%	19.3–31.8%	15.3%	11.6–19%	0.021
T lymphocytes (/µL)	2455	1482–3939 *	2788	2099–3478	0.07
(%)	57.4%	43.2–71.6%	58.3%	44.7–60% *	0.77
T Helper (T CD4+) (/µL)	1159	675–1643	1831	1282–2381	0.071
(%)	40.15%	31.7–48.6%	33.4%	25.1–42%	0.391
T CD4+ naïve (/µL)	1513	1037–1989	845	450–1259	0.181
(%)	29.4%	23.4–35.1%	29%	23.8–34.2%	0.911
T CD4+ memory (/µL)	313	231–416 *	272	232–457 *	0.962
(%)	6.3%	5.1–0.3%	7.2%	5.6–8.4%	0.62
T Cytotoxic (T CD8+) (/µL)	594	433–1054	1241	638–1.844	0.572
(%)	14.8%	9–26.5%	16.3%	13.2–19.4%	0.682
T CD8+ naïve (/µL)	476	343–811	560	369–751	0.652
(%)	14.3%	8.8–19.7%	12.3%	9.7–13.2% *	0.772
T CD8+ memory (/µL)	78	48–132 *	148	86–211	0.662
(%)	2.6%	1.3–4%	3.2%	2–4.5%	0.651
T regulatory (/µL)	2152	1763–2540	2471	2258–3120 *	0.042
(%)	65.8%	36.1–75.6% *	61.8%	25.1–79.2% *	0.515

* data not normally distributed.

**Table 5 vaccines-08-00602-t005:** Inflammatory cytokines during the febrile (acute phase) and afebrile (convalescence) phases.

Cytokine	Acute Phase	CI or IR	Convalescence	CI or IR	*p*
IL-6 (pg/mL)	385.69	13.8–50.1 *	18	4–17 *	0.05
IL-8 (pg/mL)	1646.38	33–142 *	458	19–144 *	0.95
IL12p70 (pg/mL)	1.08	0–2 *	1	0–0 *	0.4
IL-1β (pg/mL)	7.00	0–3 *	0	0–1 *	0.26
IL-10 (pg/mL)	3.92	2.5–5 *	2	1.2–2.7	0.007
TNF α (pg/mL)	9.62	1.5–19 *	18	0–15 *	0.6

* data not normally distributed.

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
