# Peer review of "Immune Response against Adenovirus in Acute Upper Respiratory Tract Infections in Immunocompetent Children"

_vaccines, 2020, doi:10.3390/vaccines8040602_

Round 1

Reviewer 1 Report

the authors present a bloodwork analysis of underage patients that presented with adenoviral infection. The authors state many hypotheses based on the differential presence of certain cytokines and cell types in the peripheral blood. However, none of these are investigated further, so that the manuscript remains a description of changes in blood cells and cytokines in acute and convalescent phase of adenoviral disease. 

No control samples are included of uninfected patients (or even bacterially infected). Which makes determination of the importance of these variations impossible. 

the impressive literature oversight in both the introduction and discussion section is much appreciated and could serve as a review manuscript. 

The problem of misdiagnosing and consequent overuse of antibiotics in children with adenoviral infections is well-appreciated, but the authors do not present any analysis or evidence that their recording of immune mediators or typical bloodwork cell counting allows for future differential diagnosis. 

Author Response

The authors present a bloodwork analysis of underage patients that presented with adenoviral infection. The authors state many hypotheses based on the differential presence of certain cytokines and cell types in the peripheral blood. However, none of these are investigated further, so that the manuscript remains a description of changes in blood cells and cytokines in acute and convalescent phase of adenoviral disease. 

Reply:

Dear Reviewer 1, Thank you for your kind review.

The purpose of our study is to describe the inflammatory response in the case of adenoviral upper respiratory tract infections that required hospitalization in children. The paper is exploring cytokines concentrations and the main lymphocytes subgroups in order to confirm, with serum markers, if Adenovirus is able to elicit systemic inflammatory response even in the case of localized infections. This study adds a hint about which subpopulation may be the main contributor to the clinical and laboratory picture.

  • No control samples are included of uninfected patients (or even bacterially infected). Which makes determination of the importance of these variations impossible. 

Reply: Unfortunately, at this stage we could not assess a control group. We understand it would be central in a prospective study, however our research design may bring knowledge in term of a cross-sectional study. We had already highlighted this among the limitations of the study; we have now added your considerations to the section (lines 402-405).

  • the impressive literature oversight in both the introduction and discussion section is much appreciated and could serve as a review manuscript. 

Reply: Thank you.

  • The problem of misdiagnosing and consequent overuse of antibiotics in children with adenoviral infections is well-appreciated, but the authors do not present any analysis or evidence that their recording of immune mediators or typical bloodwork cell counting allows for future differential diagnosis. 

Reply: Our study is a first overview about which cytokines and lymphocyte subsets are involved in the immune response against HAdV. Future studies including viral or bacterial control groups might address the important point that you raised.

Reviewer 2 Report

Authors studied 23 immunocompetent children admitted to our Pediatric Emergency Unit with signs of acute Adenoviral AURTIs, aiming at better clarifying the biological background sustaining this clinical presentation. Pediatricians often experience leukocytosis and elevated CRP in patients with AURTIs. The importance of this study will be increased by adding data and making discussion for the following.

  1. Please make discussion about the difference in immune response between AURTIs and bacterial infections.
  2. Please make discussion about the mechanism by which elevated CRP and procalcitonin are elevated in AURTIs.
  3. It is important to analyze leukocytes, especially neutrophil fraction due to shape of nuclei, such as band-shaped nuclei neutrophil and segmented nuclei neutrophil, in order to understand the pathophysiology of infectious diseases. Did you analyze the neutrophil fraction in this study?
  4. Line 139: Study design and participants
  5. Please indicate the criteria for hospitalization.
  6. Line 159: When do you have your patients tested for Adenovirus antigen? If the patient is diagnosed at the time of consultation, inappropriate antibiotics can be avoided.
  7. Line 221: Please indicate the criteria antibiotic administration.
  8. Table 4:
  9. Please show the percentage of T-reg in the table.

Author Response

Authors studied 23 immunocompetent children admitted to our Pediatric Emergency Unit with signs of acute Adenoviral AURTIs, aiming at better clarifying the biological background sustaining this clinical presentation. Pediatricians often experience leukocytosis and elevated CRP in patients with AURTIs. The importance of this study will be increased by adding data and making discussion for the following.

Reply:

Dear Reviewer 2, thank you for your kind review.

  • Please make discussion about the difference in immune response between AURTIs and bacterial infections.

Reply: We reported data concerning bacterial infections and respiratory tract infections: in cases of bacterial lower respiratory tract infections a tendency to higher levels of inflammatory cytokines IL-6 and lower levels of IL-10 is observed when compared to respiratory viral infections. (lines 373-387)

  • Please make discussion about the mechanism by which elevated CRP and procalcitonin are elevated in AURTIs.

Reply: CRP and Procalcitonin are released in AURTI due to the systemic inflammatory response elicited by interleukins released at the site of infections. (lines 116-119)

  • It is important to analyse leukocytes, especially neutrophil fraction due to shape of nuclei, such as band-shaped nuclei neutrophil and segmented nuclei neutrophil, in order to understand the pathophysiology of infectious diseases. Did you analyse the neutrophil fraction in this study?

Reply: Unfortunately, we did not analyze neutrophil fraction due to the shape of nuclei in the reported cohort. Such data would surely add important information, however the extensive immunological characterization in terms of immune system subsets distribution and inflammatory cytokines production could partly clarify some aspects of immune system activation during AURTI. The neutrophil nuclear structure and its role in migration and the release of neutrophil extracellular traps could be an excellent point to deepen for further studies about AURTI in human or in other infections leading to a significant inflammatory response. We added a short paragraph in the limitations of the study (lines 400-402)

  • Line 139: Study design and participants
  • Please indicate the criteria for hospitalization.

Reply: Criteria for hospitalization have been added. Briefly, the need of supportive therapy and the exclusion of serious bacterial infections or inflammatory disease were the drivers of hospitalization. (lines 358-361)

  • Line 159: When do you have your patients tested for Adenovirus antigen? If the patient is diagnosed at the time of consultation, inappropriate antibiotics can be avoided.

Reply: Adenovirus was sampled at the time of admission, as we do not use rapid antigen detection (RAD) kits. Routinely, results took 24-48 hours depending on the method. Antibiotics were mostly given empirically before admission after consultation with the GP or pediatrician, who similarly, do not use RAD routinely. We further clarified it in Materials and Methods section (lines 122-135)

  • Line 221: Please indicate the criteria antibiotic administration.

Reply: Briefly, antibiotic therapy was continued if started at home and discontinued as microbiolgical evidence of Adenovirus was available. In-hospital, antibiotics were given empirically in the evidence of an ill-appearing child with laboratory tests pointing to a bacterial infection. (lines 192-198)

  • Table 4:
  • Please show the percentage of T-reg in the table.

Reply: Percentage of t-reg have been added to table 4.

Reviewer 3 Report

The manuscript by Biserni et al., aims to better clarify the biological background sustaining the clinical presentation. Overall the study seems to be  without proper controls and far less N number of individuals.

  1. The introduction needs to be significantly reduced as it often felt like reading a review article.
  2. 23 individuals were assessed which is far less a set to draw any conclusions from

Author Response

The manuscript by Biserni et al., aims to better clarify the biological background sustaining the clinical presentation. Overall the study seems to be without proper controls and far less N number of individuals.

Reply:Dear Reviewer 3, thank you for your kind review.

We would like to specify that our aim was to conduct a cross-sectional study, and not a prospective study, as we had previously but erroneously stated (corrected, see line 99-104). This might have generated a misunderstanding about the study sample size. In this kind of study, it is possible to calculate prevalence of a disease and describe characteristics of the chosen population, based on a small sample. The work is exploring cytokines concentrations and the main lymphocytes subgroups in order to observe, with serum markers, the inflammatory response elicited by adenovirus in the case of localized infections. In particular, our work may open the field to future researches aiming at verifying which subpopulation may be the main contributor to the clinical and laboratory picture. Unfortunately, at this stage we could not assess a control group. We understand it would be central in a prospective study; we highlighted this among the limitations of the study (lines 402-405). A scientific English native-language speaker provided language review.

  • The introduction needs to be significantly reduced as it often felt like reading a review article.

Reply: We reduced the introduction and made it more straight-to-the-point as you gently suggested

  • 23 individuals were assessed which is far less a set to draw any conclusions from

Reply: We explained how the sample was calculated based on a power of 80%, specifying the type of errors. We included patients also considering the drop-off due to technical limits and ethical concerns.

Round 2

Reviewer 1 Report

/

Reviewer 2 Report

The authors responded to the peer-reviewed comments one by one.

Reviewer 3 Report

The manuscript in its current form can be accepted for publication.